# The Importance of Arterial Stiffness Assessment in Patients with Familial Hypercholesterolemia

**DOI:** 10.3390/jcm11102872

**Published:** 2022-05-19

**Authors:** Beáta Kovács, Orsolya Cseprekál, Ágnes Diószegi, Szabolcs Lengyel, László Maroda, György Paragh, Mariann Harangi, Dénes Páll

**Affiliations:** 1Division of Metabolism, Institute of Internal Medicine, Faculty of Medicine, University of Debrecen, 4032 Debrecen, Hungary; kovibea01@gmail.com (B.K.); dioszegi.agnes@med.unideb.hu (Á.D.); szabolcs.lengyel@med.unideb.hu (S.L.); paragh@belklinika.com (G.P.); pall.denes@unideb.hu (D.P.); 2Department of Surgery, Transplantation and Gastroenterology, Semmelweis University, 1085 Budapest, Hungary; cseprekal.orsolya@med.semmelweis-univ.hu; 3Department of Medical Clinical Pharmacology, Faculty of Medicine, University of Debrecen, 4032 Debrecen, Hungary; maroda.laszlo@unideb.hu

**Keywords:** familial hypercholesterolemia, selective LDL apheresis, PCSK9 inhibitor monoclonal antibody, arterial stiffness

## Abstract

Cardiovascular diseases are still the leading cause of mortality due to increased atherosclerosis worldwide. In the background of accelerated atherosclerosis, the most important risk factors include hypertension, age, male gender, hereditary predisposition, diabetes, obesity, smoking and lipid metabolism disorder. Arterial stiffness is a firmly established, independent predictor of cardiovascular risk. Patients with familial hypercholesterolemia are at very high cardiovascular risk. Non-invasive measurement of arterial stiffness is suitable for screening vascular dysfunction at subclinical stage in this severe inherited disorder. Some former studies found stiffer arteries in patients with familial hypercholesterolemia compared to healthy controls, while statin treatment has a beneficial effect on it. If conventional drug therapy fails in patients with severe familial hypercholesterolemia, PCSK9 inhibitor therapy should be administered; if these agents are not available, performing selective LDL apheresis could be considered. The impact of recent therapeutic approaches on vascular stiffness is not widely studied yet, even though the degree of accelerated athero and arteriosclerosis correlates with cardiovascular risk. The authors provide an overview of the diagnosis of familial hypercholesterolemia and the findings of studies on arterial dysfunction in patients with familial hypercholesterolemia, in addition to presenting the latest therapeutic options and their effects on arterial elasticity parameters.

## 1. Introduction

Cardiovascular diseases are still the leading cause of mortality worldwide, which is primarily due to increased atherosclerosis. In the background, the most important risk factors include hypertension, age, male gender, hereditary predisposition, diabetes, obesity, smoking and lipid metabolism disorders [1]. The proportion of people with high cholesterol levels is the highest in Europe in a worldwide comparison: it affects one in every two people [2,3]. Lipid metabolism is a complex process, with several known diseases that can significantly influence the lipid parameters of the serum, including cholesterol levels (secondary hypercholesterolemia). In two thirds of the cases, genetic factors are responsible for a pathological functioning of lipid metabolism (primary hypercholesterolemia). Polygenic forms are the most common, stemming from a cumulative effect of minor genetic deviances and gene variants, but more rarely, severe lipid metabolism disorders can be attributed to the mutations of individual genes [4].

## 2. Genotype and Phenotype of Familial Hypercholesterolemia

Diagnosing primary hypercholesterolemia is based on hereditary features as well as the lack of secondary factors. The group of patients suffering from familial hypercholesterolemia (FH) is prominent due to the prevalence of the disease and the severity of related cardiovascular complications. The disease, which was formerly described as a result of an inactivating mutation of the low-density lipoprotein (LDL) receptor (LDLR), is today accounted for as familial hypercholesterolemia syndrome, including both the classic form of the disease as well as a form caused by the loss-of-function mutation of apolipoprotein B100 (ApoB100), formerly termed familial defective apoB syndrome, in addition to other severe types of hypercholesterolemia similarly exhibiting autosomal dominant inheritance, some of which have been found to be related to gain-of-function mutations of the proprotein convertase subtilisin/kexin type 9 (PCSK9) gene [5]. In 2014, STAP1 (signal transducing adaptor family member 1) was reported as a novel FH candidate gene [6]. However, functional validation studies have not been reported, and possible mechanisms by which STAP1 could influence plasma lipid levels have not been explored. Indeed, no marked changes have been demonstrated in plasma lipid profiles of carriers of STAP1 variants compare to controls as well as in a mouse model [7]. Moreover, global loss of Stap1 in mice did not result in an abnormal lipid phenotype [8]. Accordingly, following these negative findings, the combined studies exclude STAP1 as an FH gene [9]. The rather rare autosomal recessive form is caused by the mutation of LDL receptor adaptor protein 1 (LDLRAP1) [10] (Table 1).

It must be noted that in clinically diagnosed FH patients without mutations in the classical genes, elevated LDL-C levels might have a polygenic cause. Such patients often carry a cluster of common polymorphisms affecting several loci associated with markedly raised LDL-C levels, comparable to those observed in patients carrying FH-causative mutations. Even in patients with monogenic FH, a polygenic contribution may subsist, contributing to the variable phenotypic expression [11]. Both monogenic FH and polygenic hypercholesterolemia are found to be associated with greater risk of cardiovascular disease (CVD) than hypercholesterolemia without a known genetic cause, with monogenic FH associated with the greatest risk [12].

### 2.1. Heterozygous FH

Genetically, the less severe heterozygous form of FH is more common than it has been previously believed. Its prevalence in the European population is currently estimated at approximately 1:300, but it may reach 1:200 [13]. According to the findings of a Hungarian research project conducted a few years ago on a large patient population, the prevalence of FH in Hungary, similarly to several European countries, is around 1:340 [14]. The study found that in addition to a substantial rise in total cholesterol and LDL-cholesterol (LDL-C) levels, the level of triglyceride did not increase in general, and the level of high-density lipoprotein-cholesterol (HDL-C) approximated the upper normal domain. Lipid abnormalities are acquired already in childhood, so screening the entire population for cholesterol levels no later than at age 8–11 has been proposed [13,14,15].

The LDL-C level is also significantly high in heterozygous cases between 4.9 and 11.6 mmol/L. As a result of the early and extremely high total cholesterol, atherosclerosis is already pronounced in childhood, and without treatment it may well lead to coronary artery disease before the patient is 35, but other vascular diseases—cerebrovascular disease or lower extremity arterial disease—have a greater risk. There is also a risk of developing tendonous xanthomata above extensor tendons, xanthelasmata on the eyelids, and corneal arcus on the iris. These are conspicuous deviations, which may help in diagnosing the disease.

Diagnosing FH is assisted by the Dutch Lipid Clinic Network criteria, which rests on three pillars: clinical symptoms, deviations in laboratory findings, and—in the case of ambiguous laboratory and clinical symptoms—a genetic test [16] (Table 2). Treatment relies on the efficient administering of statins in high doses, which in most cases is complemented by ezetimibe. As the patients are at high or extremely high cardiovascular risk, in order to reach lipid targets, it may become necessary to introduce PCSK9-inhibitor monoclonal antibodies, too [17]. Inclisiran, a small interfering RNA (siRNA) therapy, is another non-statin medication available for cholesterol management [18]. Cardiovascular screening for patients and their family members is obligatory in the case of FH [19].

### 2.2. Homozygous FH

The homozygous form of FH is a rare but rather severe disease, characterized by extreme increased LDL-C (above 12.9 mmol/L) from birth. Progressive atherosclerosis causes myocardial infarction and vascular complications in other parts of the body even in childhood, but cases of aortic valve stenosis as well as subvalvular aortic stenosis have also been recorded. Prevalence is 1:1,000,000, but due to premature mortality it is supposedly higher [20].

To examine the interconnections between the genotype and the phenotype, we need to pose the question whether the severe clinical phenotype is unequivocally tied to the homozygous mutation of a single gene. True homozygous cases, in which the same pathogenic mutation occurs in identical candidate genes in both DNA strands, result in a severe clinical disease, but similarly severe clinical diseases can be caused by compound and double heterozygous forms, too, in which a pathogenic mutation develops in different sections of the given gene, or in two different candidate genes. Therefore, when a homozygous form is suspected, it is reasonable to perform a genetic test.

## 3. Treatments of Familial Hypercholesterolemia

### 3.1. Pharmacological Treatment

The basis of the treatment is administering high-dose, intensive statin, then, if that should prove insufficient, it is combined with ezetimibe. If the LDL-C target is not accomplished, PCSK9-inhibitor treatment should be provided as a third stage [19]. In the case of homozygous as well as severe homozygous FH, the impact of statin, ezetimibe and PCSK9-inhibitors is for the most part inefficient. In the case of homozygous FH, the administering of lomitapid, a microsomal transfer protein inhibitor [21], or mipomersen, an ApoB100 synthesis inhibitor antisense oligonucleotide [22] is recommended.

In addition to these, in the case of severe heterozygous and homozygous cases, it may become necessary to provide selective LDL apheresis treatment.

### 3.2. Selective LDL Apheresis Treatment

LDL apheresis treatment involves the removal of atherogenic lipid fractions by a selective extracorporeal procedure. During apheresis, the particles containing ApoB100 are removed selectively, which can acutely reduce total and LDL-C levels by a further 50–75% beyond pharmacological treatment. In addition to that, it reduces the level of lipoprotein (a) (Lp(a)) [23], known as an individual cardiovascular risk factor also in FH patients [24,25], as well as very low-density lipoprotein levels (VLDL), also containing ApoB, by 70% [26]. During apheresis, the level of protein-like components causing cardiovascular diseases is decreased in the serum; thus, the treatment has an evidently beneficial anti-atherosclerotic impact [27]. The serum levels of inflammatory cytokines and oxidative stress are reduced [28]. Vasodilation increases, and beneficial hemorheological changes are affected [29,30].

In accordance with international recommendations (by the Food and Drug Administration), in Europe it is recommended to perform LDL apheresis in the following patient groups: *functional FH homozygote, LDL-C > 13 mmol/L, functional FH heterozygote, LDL-C > 7.8 mmol/L, functional FH heterozygote with documented ischemic heart disease and LDL-C > 5.2 mmol/L.* Alongside the recommended diet and lipid lowering pharmacological treatment at the maximum tolerated dose, LDL-C levels must exceed the specified target throughout the course of 6 months. Alternative indication is provided by findings that show *below 40% LDL-C decrease in heterozygous FH patients* alongside lipid lowering pharmacological treatment at the maximum tolerated dose. Further indication is provided by an over 60 mg/dL Lp(a) level at documented ischemic heart disease, with over 4 mmol/L LDL-C level despite the administered pharmacological treatment. The above treatments have yielded favorable results, but due to limited funding the number of patients systematically treated in Europe is unfortunately still low [31].

## 4. Significance of Cardiovascular Risk Assessment

Cardiovascular morbidity and mortality are significantly enhanced in patients with different types of FH; therefore; they all would require individual cardiovascular risk assessment at the earliest possible stage of the disease course [32]. Not all FH patients are at the same cardiovascular risk [33]. The main predictors of morbidity and mortality in this patient population are known to be the age HDL-s, gender, hypertension, and smoking, which all together make up the Montreal FH risk prediction score [34]. Those individual biomarkers do not provide direct information about the hard outcomes and moreover, their cumulative effect could only be declared only if hard endpoints have already occurred. Biomarkers and comorbidities result in an intermediate state of decreased arterial elasticity as a surrogate endpoint of cardiovascular hard outcome in this patient population [35]. Signs of accelerated athero- and arteriosclerosis may occur in early childhood in special types of FH, which refer to the hazard of later functional and structural arterial damage. AHA for instance classified homozygous FH as a Tier I risk group from early childhood [36].

Early risk assessment and patient education are crucial factors to prevent later fatal outcomes. Arterial stiffness measurement contributes to understanding cardiovascular morbidity and mortality risk beyond traditional risk factors or blood pressure measurement [37]. There are several methods to assess arterial stiffness. Due to the fact that there are plenty of measurement tools to assess central and peripheral vascular elasticity, none of them has been approved as a standard method to routinely measure stiffness as a surrogate marker [38]. Nonetheless, ESH suggests the non-invasive measurement of central PWV as the gold standard method to assess preclinical organ damage in patients at high cardiovascular risk [39]. Some studies proved that beyond traditional risk assessment tools, and it may offer additional value and refinement of strategies applied thus far.

Measuring arterial stiffness as a surrogate marker is of paramount importance and will further help lengthen the survival of FH patients.

## 5. Arterial Stiffness

Suitable non-invasive methods in early stages of atherosclerosis and related artery wall disorders include measuring arterial stiffness. Increased arterial wall stiffness is a result of complex structural changes in the tunica media of the great arteries and of their increased and progressive calcification. Severe arterial stiffness precedes the development of atherosclerosis, which then causes the symptoms. Functional parameters that are independent predictors of cardiovascular diseases include applied pulse wave velocity (PWV) and augmentation index (Aix), which are suitable for both screening and monitoring the efficiency of the treatment [40]. In the past years, the 24-h monitoring of stiffness has become increasingly widespread. Internationally these measurements are most often performed by devices using the oscillometric method, such as Vasotens^®^ (BPLab GmBH Schwalbach am Taunus, Hessen, Germany) and Mobil-O-Graph^®^ (IEM, Stolberg, Germany). Arterial stiffness parameters—pulse wave velocity, augmentation index, central blood pressure—are recorded throughout 24 h with the help of an upper arm blood pressure monitor. The development of early-onset artery wall dysfunction with increased cardiovascular risk was first diagnosed in diseases with chronic inflammation and lipid metabolism disorder. Artery wall dysfunction and related chronic inflammation involve a change in the marker levels of several inflammatory proteins and other serum markers [41].

## 6. Relationship between Cholesterol and Arterial Stiffness

The link between serum cholesterol level and arterial stiffness may be explained by several potential mechanisms [42]. The more obvious and probably the most important is the development of atherosclerosis, which has been consistently associated with increased arterial stiffness in subjects with and without severe hypercholesterolemia. However, cholesterol and especially oxidatively modified LDL (oxLDL) have further, non-atheromatous effects on the arterial wall, leading to arterial stiffening. OxLDL promotes peroxynitrite formation and increased oxidative stress, which may lead to the direct damage of elastin, the main elastic element of the arterial wall [43]. Furthermore, oxLDL has pro-inflammatory effects characterized by increased serum levels of C-reactive protein (CRP), which was associated with arterial stiffness in apparently healthy individuals [44]. Inflammatory cytokines enhance the expression of some inducible enzymes, such as matrix metalloproteinase-9 (MMP-9), that may damage the structural components of the arterial wall. MMP-9 is a gelatinase secreted by immigrating inflammatory cells of the vascular wall, capable of digesting elastin leading to the remodeling of the arterial wall [45]. Local inflammation and inflammatory lipids may also promote calcium deposition in the medial elastic fibers, resulting in arterial calcification [46]. In addition to structural changes, hypercholesterolemia induces functional dysregulation of the vascular endothelium found to be associated with arterial stiffness. High serum levels of cholesterol are significantly associated with reduced bioavailability of nitric oxide, impaired L-arginine/nitric oxide pathway and increased asymmetric dimethyl arginine production, leading to impaired endothelial vasodilatation [47]. On the other hand, vascular dysfunction may also be caused by the overproduction of vasoconstrictor agents, including endothelin-1 [48].

The relationship between serum lipid levels and arterial stiffness have been examined in several former studies. Most of them demonstrated a significant positive relationship between large artery stiffness and total or LDL-cholesterol [49,50,51,52]. It must be noted that many of these studies included a relatively low number of patients with other cardiovascular risk factors. Therefore, careful interpretation of the data is essential. FH represents an extreme form of total and LDL cholesterol elevation; therefore, it may serve as an excellent model to prove the link between LDL cholesterol and arterial stiffness.

## 7. Assessing Arterial Stiffness in Familial Hypercholesterolemia

Patients with familial hypercholesterolemia have an extremely high risk of atherosclerosis and early-onset vascular ageing, which can be measured by the increase of arterial stiffness. This is due to high cholesterol levels, including the presence of high serum LDL-C values as well as high Lp(a), more prevalent than in the general population, and higher levels of oxidized LDL and chronic artery wall inflammation due to increased oxidative stress [53]. In the case of FH, a low-fat diet and a conventional lipid lowering treatment have limited efficiency and due to the already existing arterial complication, often it is not possible to do physical exercise to the desired effect. All of these may be exacerbated by conventional cardiovascular risk factors, such as age, excess weight, diabetes, hypertension, and smoking. These may be accompanied by other unfavorable genetic factors as well [54] (Figure 1).

As a result of accelerated atherosclerosis, sooner or later all patients develop coronary stenosis; however, its extent and the severity of the resulting clinical symptoms have a broader spectrum. Even though the injurious effects of arterial stiffness on the population of non-FH patients have been registered by several studies, the available data on the clinical impact of arterial stiffness on FH patients is insufficient.

In a small cross-sectional comparative study, brachial-ankle pulse wave velocity (baPWV) was measured in 35 heterozygous FH subjects and 17 healthy control subjects. Although baPWV disi not differ significantly between FH patients and controls (12.5 ± 2.9 vs. 11.9 ± 2.3 m/s), among FH patients, the baPWV and carotid IMT were higher in cases with high cholesterol burden than those without. Similarly, the baPWV and carotid IMT were also higher in cases with elevated hs-CRP than those without [55].

In a former case control study of 22 patients with FH and matched healthy controls, PWV values were compared before and after lipoprotein apheresis (LA) treatment. Baseline PWV was similar between the two groups (controls 8.2  ±  0.9  m/s vs. FH 7.7  ±  0.8  m/s, *p*  =  0.12). Moreover, baseline PWV did not change following LA (pre 8.8  ±  1.2  m/s vs. post 9.2  ±  1.2  m/s, *p*  =  0.19) [56].

Another study involved 60 patients without documented cardiovascular events and clinical symptoms of cardiovascular diseases: 21 patients with elevated plasma LDL-C levels and genetically confirmed FH, 19 patients with elevated LDL-C levels and without FH mutations and 20 healthy controls. In each patient, echo-tracking and photoplethysmography were used to assess the parameters of arterial stiffness. They found that arterial stiffness parameters were similar between the groups [57].

A study conducted on the population of 125 FH patients as per the guidelines displayed significantly higher Aix values in comparison to those of a control group of identical age and gender (9.6 ± 17.2 vs. 2.6 ± 10.3%; *p* = 0.011), based on which the measuring of Aix value is recommended in patient tracking [58].

In a study conducted on 66 untreated FH patients and 57 first-degree non-FH relatives, when measuring carotid β-stiffness index and carotid-femoral PWV it was found that while FH patients’ β-index (6.3 (4.8–8.2) vs. 5.2 (4.2–6.4); *p =* 0.005) and local PWV values (5.4 (4.5–6.4) vs. 4.7 (4.2–5.4) m/s; *p =* 0.005) were significantly higher than in the case of their non-FH relatives, there was no substantial deviation in carotid-femoral PWV values (6.76 (7.0–7.92 vs. 6.48 (6.16–7.12) m/s; *p =* 0.138). Based on all the above, the measurement of carotid arterial stiffness, especially in the case of younger patients, may indicate the extent of calcification sooner than does arterial stiffness of the aorta [59].

A Japanese group of researchers recorded changes in brachial-tibial pulse wave velocity (baPWV) as well as the development of coronary artery disease in 245 medicated FH individuals. The patients were selected on the basis of clinical criteria for FH specified by the Japan Atherosclerosis Society. According to these, two out of three clinical criteria need to be met for a diagnosis of FH, namely, LDL-C ≥ 180 mg/dL, the presence of tendonous xanthoma or xanthoma tuberosum, as well as an FH-positive family history or early-onset CAD diagnosed in second-degree relatives. Cardiovascular risk factors (age, male gender, hypertension, diabetes, smoking) have been assessed as well as deviances in lipid parameters (total cholesterol, triglycerides, HDL) and the presence of CAD. In the latter case, the diagnosis was established on the basis of coronary CT angiography by taking into account only over 50% stenosis of the main coronary arteries. Measurement of brachial-tibial pulse wave velocity was performed with a Colin VP-1000, Omron^®^ device. The goal of the study was to establish a connection between arterial stiffness and the risk of CAD in the given population. The findings proved that in the case of FH arterial stiffness, including baPWV as a biomarker indicating high cardiovascular risk, showed correlation with CAD [60].

Parameters of arterial stiffness as well as aortic root thickness by cardiac MRI have also been tested on heterozygous FH children. Testing 33 children aged 7–18, it was found that in comparison to a non-FH group of identical age, the PWV values of FH children were significantly higher (4.5 ± 0.8 vs. 3.5 ± 0.3 m/s; *p* < 0.001), and the wall thickness of the ascending aorta was higher (1.37 ± 0.18 vs. 1.3 ± 0.02 mm; *p* < 0.05), which suggests the importance of early statin treatment [61].

However, some further studies in children and young patients with FH with low patient numbers could not demonstrated significant differences compared to control subjects [35,62,63].

Indeed, a recent meta-analysis of 8 studies involving 317 patients with FH and 244 non-FH individuals did not suggest a significantly altered PWV in FH patients versus controls, although the authors admit that different scores for FH diagnosis as well as different methods for PWV estimation were used in different studies included and there was a lack of information about the duration and type of lipid-lowering therapy [64].

Taken together, larger studies evaluating PWV in FH patients compared with controls in order to elucidate the impact of FH on arterial stiffness as measured by PWV are definitely needed. A very recent position paper of the Associations of Preventive Paediatrics of Serbia, Mighty Medic and International Lipid Expert Panel focusing on risk assessment and clinical management of children and adolescents with heterozygous FH stated that depending on the availability of noninvasive equipment for PWV measurement and staff experience, it would be clinically meaningful to perform PWV measurements in all children with FH and evaluate their changes over time. PWV values above 97th could be a possible guide for treatment initiation in ambiguous clinical cases (Level B evidence). The paper prefers oscillometric devices; it suggests the usage of Mobile-O-Graph device due to the simplicity of measurement [65].

Whether the parameters for establishing arterial stiffness can be replaced by measuring inflammatory parameters of atherosclerosis is an important question. According to the findings of a study conducted on 89 FH patients and a control group of 31, PWV values were higher in FH patients (*p* < 0.05), but no significant connections to serum inflammatory parameters have been found (C-reactive proteins and white blood count) [66].

## 8. Effect of Traditional Oral Lipid-Lowering Treatment on Arterial Stiffness

Several human studies have investigated the effect of HMG-CoA reductase inhibitor statins on arterial stiffness. It is well documented that in addition to cholesterol reduction, a number of other pleiotropic effects have been described with statins that may improve atherogenesis independently of cholesterol reduction, including their anti-inflammatory and antioxidant effects, leading to improved endothelial function [67,68]. With the exception of one short-term, cross-over study, which could not find any impact of pravastatin on carotid, brachial and femoral artery stiffness [69], most of these former studies reported significant reduction in stiffness parameters after higher doses of atorvastatin and cerivastatin treatment [70,71,72,73]. Even a long-term, but low-dose pravastatin treatment could improve the aortic pulse wave velocity, especially in patients with the greatest lowering of cholesterol [74]. It must be noted that most statins had significant adjuvant effects on peripheral systolic blood pressure [72,75]. On the contrary, 20-week treatment with statins (rosuvastatin or atorvastatin) combined with regular exercise significantly improved exercise capacity and brachial artery PWV, but had no effect on blood pressure [76]. In patients with coronary artery disease compared with simvastatin/ezetimibe, rosuvastatin was found to more effectively improve arterial wall stiffness [77]. Although the beneficial effect of statin on stiffness parameters in various patient populations are well established, to date, their effects on stiffness parameters in FH patients has not been investigated.

Recently, the impact of PCSK9 plasma levels on mechanical vascular impairment was verified [78]. The exact mechanism is not fully clarified, but PCSK9 might promote atherogenesis by stimulating oxidative stress and the production of proinflammatory cytokine production of the atherosclerotic lesions [79]. It must be highlighted that statins, especially the lipophilic agents significantly increase the circulating level of PCSK9. Furthermore, statin-induced PCSK9 increase may limit the absolute magnitude of statin LDL-C lowering effect, by limiting the statin-driven LDLR upregulation [80].

## 9. Changes in Arterial Stiffness and PCSK9-Inhibitor Monoclonal Antibody Treatment

In the past few years, new product groups have appeared in the range of FH therapeutic medicines. Breakthrough products included the aforementioned ApoB synthesis inhibitors, microsomal transfer protein inhibitors and especially PCSK9-inhibitors. Currently, the efficiency, side effect profiles and effect of PCSK9-inhibitor monoclonal antibodies on the cardiovascular endpoints are very similar. In the RUTHERFORD study, 168 heterozygous FH patients were treated with 350 mg and 420 mg evolocumab every 4 weeks alongside statin treatment. In the case of 350 mg, LDL-C decreased by 43%, in the case of 420 mg by 55%. After 12 weeks, 44% and 65% of the patients, respectively, reached the desired 1.8 mmol/L target value. Triglyceride levels decreased by 15% and 20%, respectively, while HDL-C increased by 7%, and Lp(a) fell by 23% and 32%, respectively, as a result of evolocumab treatment. In the case of administering 140 mg every 2 weeks, LDL-C decreased by 66% [81]. Therefore, the impact on total cholesterol and LDL-C is conclusive; however, the impact on Lp(a) level lags that of LDL apheresis.

Changes in lipid parameters and pulse wave velocity due to complementary treatment using PCSK9-inhibitors or ezetimibe were studied in a 6-week tracking study. The research project involved ninety-eight certified FH patients who had previously undergone other cardiovascular risk assessment. All patients had genetically certified FH. The patients had received high-dose statin (atorvastatin 40–80 mg, rosuvastatin 20–40 mg) and/or ezetimibe treatment at least 6 months prior to the commencement of the study, but still they had not reached the desired LDL-C targets. In the study, 53 patients were administered statin+ezetimibe+PCSK9-inhibitors (alirocumab 75 mg/150 mg or evolocumab 140 mg). Forty-five patients were administered ezetimibe alongside the statin already administered. PWV measurements had been taken prior to the complementary pharmacological treatment and 6 months after the optimized treatment. Measurements were conducted with the SphygmoCor CVMS^®^ device. In the case of patients in the PCSK9 group, a more significant decrease was recorded not only in LDL-C values, but also in pulse wave velocity (−51% vs. −22.8%, *p* < 0.001 and −15% vs. −8.5%, *p* < 0.01) [82].

The effect of six-month add-on PCSK9 inhibitor monoclonal antibodies on circulating PCSK9 and PWV was detected in a cohort of FH subjects. The PCSK9 plasma level was correlated with PWV at baseline. Furthermore, reduction of PCSK9 plasma level seems to be associated with a significant mechanical vascular improvement after PCSK9 inhibitor monoclonal antibody therapy. Therefore, PCSK9 could be a novel cardiovascular biomarker of the mechanical vascular homeostasis through lipid and non-lipid pathways, and it could identify subjects at high CVD risk with a limited LDL-C lowering benefit after high-intensity statin therapy in FH [78].

## 10. The Impact of LDL Apheresis Treatment on Vascular Parameters

The impact of LDL apheresis treatment on arterial stiffness is the least documented in spite of the fact that in acute cases this extracorporeal procedure yields the most substantial metabolic and hemodynamic changes. Professional literature on the subject is also rather scarce. A German research team examined the impact of lipoprotein apheresis treatment on the parameters of endothelial function (circulating endothelial cells, circulating endothelial progenitor cells, flow-mediated vasodilation, microalbuminuria) as well as left ventricular ejection fraction and changes in homocystein levels. Heterozygous FH patients were examined: 21 patients were administered statin at the maximum tolerated dose, while 8 patients proved to be statin intolerant. Direct adsorption of lipoproteins (DALI) was provided on a weekly basis. Primarily in the case of the statin intolerant patient’s immediate improvement was recorded in vascular contractility even after a single treatment. Regular apheresis throughout a course of 6 months clearly had a favorable effect on the metabolic parameters under survey and improved endothelial function, which is one of the key causes of clinical improvement [83].

## 11. Conclusions

In familial hypercholesterolemia, complex molecular and hemodynamic changes are involved in the development of cardiovascular complications. Even though an increasing amount of clinical data is available, the exact role of serum cholesterol in the changing elasticity of different arterial sections is still not clear. Several combined pharmacological treatments are available for remedying metabolic deviations, but in clinically severe cases, complex pharmacological and non-medicinal treatments, such as selective LDL apheresis, can be used jointly. Similarly, little is known about changes in arterial stiffness induced by new generations of medicines, such as PCSK9-inhibitor monoclonal antibodies, siRNAs and selective LDL apheresis treatments. Atherosclerosis and related artery wall dysfunction can be screened for by non-invasive arterial stiffness measurements. Today oscillometric ABPM devices are available for performing a 24-h measurement of arterial stiffness; these are used primarily in scientific research but not widespread in clinical practice. Since these devices are suitable for measuring biomarkers indicating high cardiovascular risk, such measurements may contribute to screening especially high-risk patients with familial hypercholesterolemia and to improving the efficiency of their treatment.

## Figures and Tables

**Figure 1 jcm-11-02872-f001:**
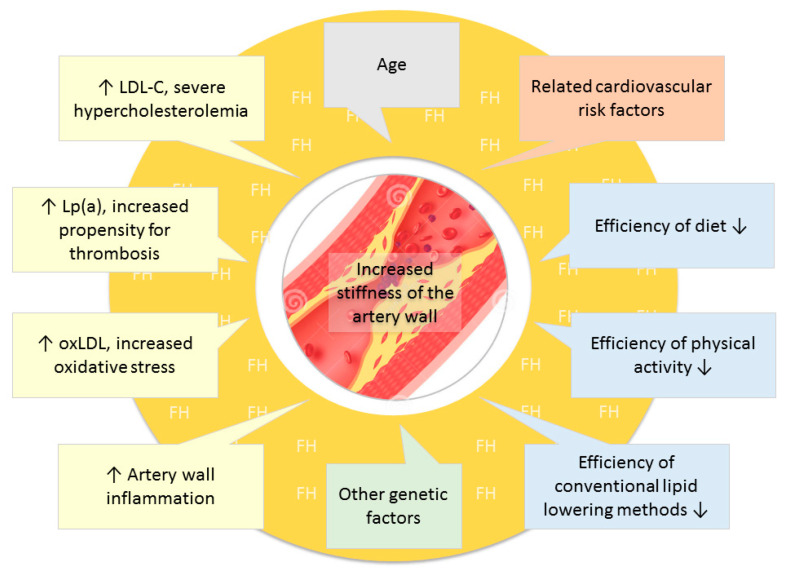
Factors leading to increased vascular stiffness in familial hypercholesterolemia. LDL-C: low density lipoprotein-cholesterol, Lp(a): lipoprotein (a); oxLDL: oxidized low-density lipoprotein.

**Table 1 jcm-11-02872-t001:** Genetic deviations in the background of FH syndrome [4].

Inheritance	Chromosome	Gene	Name	Prevalence
**Autosomal dominant**	19p13	LDLR	Familial hypercholesterolemia (FH)	60–80%
	2p24–p23	ApoB100	Familial defective ApoB syndrome (FDB)/(FH2)	1–5%
	1p32	PCSK9	PCSK9 gain-of-function (FH3)	0–3%
		several genes	Polygenic forms	20–40%
**Autosomal recessive**	1p35	LDLRAP1	Autosomal recessive hypercholesterolemia	rare

ApoB100: apolipoprotein B100; FH: familial hypercholesterolemia; FDB: familiar defective ApoB; LDLR: low-density lipoprotein receptor; LDLRAP1: LDLR adaptor protein 1; PCSK9: proprotein convertase subtilisin/kexin type 9.

**Table 2 jcm-11-02872-t002:** Dutch Lipid Clinic Network diagnostic criteria [16].

**Family History**	**Earl-Onset CAD or PAD First-Degree Relative**	**1 point**
	**Presence of Xanthomata or Corneal Arcus in First-Degree Relative**	**2 points**
**Clinical history**	CAD in women under 60, in men under 55	2 points
	stroke or PAD in women under 60, in men under 55	1 point
**Physical examination**	presence of tendonous xanthomata at any age	6 points
	presence of corneal arcus under 45	4 points
**Laboratory tests**	LDL > 8.5 mmol/L	8 points
	LDL 6.5–8.4 mmol/L	5 points
	LDL 5.0–6.4 mmol/L	3 points
	LDL 4.0–4.9 mmol/L	1 point
	note: HDL and TG levels norm.	
**DNA analysis**	detectable mutation in the LDL receptor gene	8 points
**The diagnosis is verified if score is higher than 8 points**		
**The diagnosis is probable: 6–8 points**		
**The diagnosis is possible: 3–5 points**		

CAD: coronary artery disease; HDL: high-density lipoprotein; LDL: low-density lipoprotein; PAD: peripheral artery disease; TG: triglyceride.

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
