# Peer review of "The Importance of Arterial Stiffness Assessment in Patients with Familial Hypercholesterolemia"

_jcm, 2022, doi:10.3390/jcm11102872_

Round 1
Reviewer 1 Report
This article aims to describe the problem of familial hypercholesterolemia with a focus on arterial stiffness. Familial hypercholesterolemia is one of the most common monogenic diseases. Patients with familial hypercholesterolemia are at very high cardiovascular risk. Arterial stiffness is an independent predictor cardiovascular risk. The authors presented a review on this topic. The topic of the work is relevant. I have one major comment.
Major Comment:
Lines 54-57 and Table 1. The STAP1 gene is shown to be associated with the development of FH. Recent work has refuted this connection (see: Hegele RA, Knowles JW, Horton JD. Delisting STAP1: The Rise and Fall of a Putative Hypercholesterolemia Gene. Arterioscler Thromb Vasc Biol. 2020 Apr;40(4):847-849. doi: 10.1161/ATVBAHA.120.314006.; Loaiza N, Hartgers ML, Reeskamp LF, Balder JW, Rimbert A, Bazioti V, Wolters JC, Winkelmeijer M, Jansen HPG, Dallinga-Thie GM, Volta A, Huijkman N, Smit M, Kloosterhuis N, Koster M, Svendsen AF, van de Sluis B, Hovingh GK, Grefhorst A, Kuivenhoven JA. Taking One Step Back in Familial Hypercholesterolemia: STAP1 Does Not Alter Plasma LDL (Low-Density Lipoprotein) Cholesterol in Mice and Humans. Arterioscler Thromb Vasc Biol. 2020 Apr;40(4):973-985. doi: 10.1161/ATVBAHA.119.313470.; Kanuri B, Fong V, Haller A, Hui DY, Patel SB. Mice lacking global Stap1 expression do not manifest hypercholesterolemia. BMC Med Genet. 2020 Nov 23;21(1):234. doi: 10.1186/s12881-020-01176-x.;)
This gene should be excluded.
Author Response
Response to Reviewer 1
Thank you for the review and the valuable comment on our paper.
The reviewer is right. The role of STAP1 gene variations in the development of FH is a matter of serious debate. In 2014, Fouchier and colleagues discovered that variants of the STAP1 gene were associated with the FH phenotype in a linkage analysis of a family with FH4 and suggested robust support for genetic causality (Fouchier). Several subsequent studies reported STAP1 gene variants observed in such FH patients also manifested significant cardiovascular events (Amor-Salamanca, Cao, Braenne). However, others found no association between STAP1 variants and hypercholesterolemia (Danyel, Sanchez-Hernandez, Pirillo, Iacocca). Moreover, an analysis of seven families with FH phenotype failed to observe the co-segregation of four rare predicted pathogenic variants of STAP1 (Lamiquiz-Moneo). Loaiza et al demonstrated no marked changes plasma lipid profiles of carriers of STAP1 variants compare to controls as well as in a mouse model (Loaiza). Finally, Kanuri et al proved that STAP1 does not alter lipid levels in Stap1 deficient mice, and they concluded that it is not causative for hyperlipidemia in patients with autosomal dominant familial hypercholesterolemia. Taken together, based on the results of combined studies in mouse models and carriers of STAP1 variants, STAP1 might not a familial hypercholesterolemia gene (Kanuri).
Therefore, we rephrased this part of the manuscript (Ln 55-65) and corrected Table 1. The STAP1 gene was deleted. The references mentioned by the reviewer are cited (Ref 7-9).
The following sentences have been added:
In 2014, STAP1 (signal transducing adaptor family member 1) was reported as a novel FH candidate gene (Fouchier). However, functional validation studies have not been reported, and possible mechanisms by which STAP1 could influence plasma lipid levels have not been explored. Indeed, no marked changes have been demonstrated in plasma lipid profiles of carriers of STAP1 variants compare to controls as well as in a mouse model (Loaiza). Moreover, global loss of Stap1 in mice did not result in an abnormal lipid phenotype (Kanuri). Accordingly, following these negative findings, the combined studies exclude STAP1 as an FH gene (Hegele).
Reviewer 2 Report
In this review, the authors adequately showed the role of arterial stiffness assessment in FH. The review is well performed, the toic is interesting and the english style is adequate. I only have minor concerns about the review:
- In the title, please change "the significance" with "the importance"
- In the Paragraph 2, please change the title to "Genotype and phenotype of familial hypercholesterolemia". Please also consider the role of polygenic FH (10.1161/JAHA.121.023668; 10.1001/jamacardio.2019.5954)
- In the Paragraph 8, a recent paper showed that PCSK9 plasma levels are associated with arterial stiffness in FH patients without ASCVD. This could demonstrate a potential role of PCSK9 plasma levels on the vascular function and remodelling and to clarify the effects of
PCSK9-i in these pathways; please consider this paper (10.3390/biom12040562) and comment this in the Paragraph 8.
Author Response
Response to Reviewer 2
Thank you for for the review, the valuable comments, and the positive opinion about our paper. Please find our answers to the questions and comments as follows.
- We modified the title according to the reviewer’s commentas follows (Ln 2):
The importance of arterial stiffness assessment in patients with familial hypercholesterolemia
- Thank you for the comment. We also modified the title of Paragraph 2 to "Genotype and phenotype of familial hypercholesterolemia" according to the reviewer’s comment (Ln 43). The role of polygenic FH is discussed (Ln 68-76), and the mentioned references (Olmastroni et al. and Trinder et al.) are cited (Ref 11, 12). The following sentences are added to the text:
It must be noted that in clinically diagnosed FH patients without mutations in the classical genes, elevated LDL‐C levels might have a polygenic cause. Such patients often carry a cluster of common polymorphisms affecting several loci associated with markedly raised LDL‐C levels, comparable to those observed in patients carrying FH‐causative mutations. Even in patients with monogenic FH, a polygenic contribution may subsist, contributing to the variable phenotypic expression (Olmastroni). Both monogenic FH and polygenic hypercholesterolemia are found to be associated with greater risk of CVD than hypercholesterolemia without a known genetic cause, with monogenic FH associated with the greatest risk (Trinder).
- Thank you for the comment. Regarding the potential role of PCSK9 plasma levels on the vascular function and remodelling and to clarify the effects of PCSK9-i in these pathways, the above-mentioned retrospective study by Toscano et al. is now mentioned and cited (Ln 354-361 and Ln 390-397, Ref 78). The following sentences are added to the text of Paragraph 8 and 9:
Recently, the impact of PCSK9 plasma levels on mechanical vascular impairment was verified (Toscano). The exact mechanism is not fully clarified, but PCSK9 might promote atherogenesis by stimulating oxidative stress and the production of proin-flammatory cytokine production of the atherosclerotic lesions (Rusicka). It must be high-lighted that statins, especially the lipophilic agents significantly increase the circulating level of PCSK9. Furthermore, statin-induced PCSK9 increase may limit the absolute magnitude of statin LDL-C lowering effect, by limiting the statin-driven LDLR up-regulation (Sahebkar).
The effect of six-month add-on PCSK9 inhibitor monoclonal antibodies on circulating PCSK9 and PWV was detected in a cohort of FH subjects. The PCSK9 plasma level was correlated with PWV at baseline. Furthermore, reduction of PCSK9 plasma level seems to be associated with a significant mechanical vascular improvement after PCSK9 inhibitor monoclonal antibody therapy. Therefore, PCSK9 could be a novel cardiovascular biomarker of the mechanical vascular homeostasis through lipid and non-lipid pathways, and it could be able to identify subjects at high CVD risk with a limited LDL-C lowering benefit after high intensity statin therapy in FH (Toscano).